# Theoretical Study of LED Operating in Noncarrier Injection Mode

**DOI:** 10.3390/nano12152532

**Published:** 2022-07-23

**Authors:** Chaoxing Wu, Kun Wang, Tailiang Guo

**Affiliations:** 1College of Physics and Information Engineering, Fuzhou University, Fuzhou 350108, China; 211110018@fzu.edu.cn; 2Fujian Science & Technology Innovation Laboratory for Optoelectronic Information of China, Fuzhou 350108, China

**Keywords:** light-emitting device, noncarrier injection mode, theoretical study, alternating voltage

## Abstract

Non-carrier injection (NCI) mode is an emerging driving mode for light-emitting diodes (LEDs) with numerous advantages. Revealing the relationship between the current and the applied alternating voltage in mathematical formulas is of great significance for understanding the working mechanism of NCI–LEDs and improving device performance. In this work, a theoretical model of the relationship between NCI–LED current and time-varying voltage is constructed. Based on the theoretical model, the real-time current is derived, which is consistent with the experimental results. Key parameters that can improve device performance are discussed, including voltage amplitude, frequency, equivalent capacitance, and LED reverse current. The theory presented here can serve as an important guidance for the rational design of the NCI–LEDs.

## 1. Introduction

Recently, micro-light-emitting diode has been emerging as a promising electronic platform for advanced applications, including ultrahigh resolution displays, visible light communications, and biocompatible lighting sources [1,2,3]. Different from conventional driving mode, noncarrier injection (NCI) mode has been demonstrated for using in micro–scale LED and nano-scale LED [4,5,6,7,8,9,10,11,12,13,14,15]. As an electrical contact between the external electrodes and the LED chip is avoided, the device structure and manufacturing process can be simplified. Therefore, NCI–LEDs have potential applications in Micro–LED displays and nano-pixel light-emitting displays (NLEDs) [16]. Contrary to common knowledge, in NCI operating mode, no electrical contact occurs between the LED and the external electrodes, and external electrons and holes do not contribute to the radiative recombination. Therefore, under an external alternating current (AC) electric field, only the inherent electrons and holes contribute to the periodical radiative recombination. The electrical and optical properties of NCI–LEDs, including current-voltage properties, current-frequency properties, brightness-frequency properties, and light-pulse splitting etc., have been measured experimentally [4,5,6,7,8,9]. However, there is still a lack of theoretical model for NCI–LEDs, which is critical for designing a compatible NCI–LED for specific applications.

As the driving voltage of NCI–LED is AC voltage, in addition to the amplitude (*V*), frequency (*f* and *ω*) and time (*t*) are also key factors affecting the device performance. Moreover, the equivalent capacitance (*C*) in NCI-LEDs is also a key parameter. In other words, the current of NCI-LED is a function of *V*, *ω*, *t*, and *C*, which is much more complicated than the conventional LED. Thus, it is important to reveal the relationship between *i* and applied voltage parameters (*V*, *ω, t*) in a mathematical formula. In this paper, mathematical equations related to the time-varying current of NCI–LED are constructed. The time-varying current derived from the mathematical equations is consistent with the experimental result. Additionally, the physics behind the time-varying behavior is clarified. Finally, key parameters that can improve device performance are discussed.

## 2. Results

The most important theoretical mode representing NCI–LED is the relationship between electroluminescence intensity and driving voltage. Considering that the electroluminescence intensity is positively related to the current flowing through the LED, we focus on the theoretical equations related to the *i–v* relationship. For NCI–LED, there are two insulating layers inserted between the electrodes and the LED; thus, the device is equivalent to an electronic system with a capacitor–LED–capacitor structure, as schematically presented in Figure 1a. From the point of view of circuit analysis, the capacitor–LED–capacitor series structure is equivalent to a capacitor–LED series structure, as shown in Figure 1b. Furthermore, it is reasonable to assume that the LED has the same *i*–*v* relationship as a PN junction. Thus, when a voltage is applied, there is a current (*i*) pass through MQWs, and positive charges (*Q*^+^) and negative charges (*Q*^−^) with the same amount accumulate on the two electrodes of the capacitor, respectively.

In order to facilitate the derivation of the equations, we assume that the applied voltage is a sinusoidal voltage of *U*_0_·sinωt, where the voltage amplitude is *U*_0_ and the angular frequency is *ω*. According to Kirchhoff Voltage Laws, the relationship between the applied voltage, the voltage drop across the capacitor (*U_C_*), and the voltage drop applied to the LED (*U*_LED_) can be given by
(1)$$U_{0}\sin\omega t=U_{C}+U_{\mathrm{LED}}$$

The *U_C_* can be given by
(2)$$U_{C}=\frac{Q}{C}=\frac{\int_{0}^{t}idt}{C}$$
where *C* is the capacitance related to the insulating layer. The relationship between *i* and *U*_LED_ can be given by [17]
(3)$$i=i_{0}\left(e^{\frac{qU_{\mathrm{LED}}}{kT}}-1\right)$$
where *i*_0_ is the inverse saturation current of a PN junction. Substituting Equations (2) and (3) into Equation (1), we can have
(4)$$U_{0}\sin\omega t=\frac{\int_{0}^{t}idt}{C}+\frac{kT}{q}\ln\left(\frac{i}{i_{0}}+1\right)$$

Differentiating both sides of the equation with respect to *t*, we can get
(5)$$U_{0}\omega \cos\omega t=\frac{i}{C}+\frac{kT}{q\left(i+i_{0}\right)}\frac{di}{dt}$$

When defining the constant of B as
(6)$$B=\frac{kT}{q}$$

Equation (5) is equivalent to
(7)$$U_{0}\omega \cos\omega t=\frac{i}{C}+\frac{B}{\left(i+i_{0}\right)}\;\frac{di}{dt}$$

Then, Equation (7) can be simplified as
(8)$$\frac{di}{dt}+\left(\frac{i_{0}}{BC}-\frac{U_{0}\omega }{B}\cos\;\;\omega t\right)i+\frac{1}{BC}i^{2}=\frac{i_{0}U_{0}\omega }{B}\cos\;\omega t$$

Note that when using nanoscale LEDs, the peak current of the circuit is typically less than 10^−5^ A. Since the current (*i*) is relatively small, Equation (8) can be approximate to
(9)$$\frac{di}{dt}+\left(\frac{i_{0}}{BC}-\frac{U_{0}\omega }{B}\cos\;\omega t\right)i=\frac{i_{0}U_{0}\omega }{B}\cos\;\omega t$$

This is a first order nonhomogeneous linear differential equation. We can define *P*(*t*) and *Q*(*t*) as
(10)$$P\left(t\right)=\frac{i_{0}}{BC}-\frac{U_{0}\omega }{B}\cos\;\omega t$$
(11)$$Q\left(t\right)=\frac{i_{0}U_{0}\omega }{B}\cos\;\omega t$$

Thus, Equation (9) can be solved analytically as
(12)$$i=Ae^{-\int_{0}^{t}P\left(t\right)dt}+e^{-\int_{0}^{t}P\left(t\right)dt}\int_{0}^{t}Q\left(t\right)e^{\int_{0}^{t}P\left(t\right)dt}dt$$
(13)$$\int_{0}^{t}P\left(t\right)dt=\frac{i_{0}}{BC}t-\frac{U_{0}}{B}\;\sin\omega t$$
where A is a constant. Then, Equation (12) can be simplified as
(14)$$i=Ae^{-(\frac{i_{0}}{BC}t-\frac{U_{0}}{B}\;\sin\omega t)}+\frac{i_{0}U_{0}\omega }{B}e^{-(\frac{i_{0}}{BC}t-\frac{U_{0}}{B}\sin\omega t)}\int_{0}^{t}\cos\omega t\cdot e^{\left(\frac{i_{0}}{BC}t-\frac{U_{0}}{B}\sin\omega t\right)}dt$$

It can be found that *i* is a variable that varies significantly with time *t*. We now only consider *i* is in a relatively stable and observable state after a sufficiently long time. Thus, we can have
(15)$$\int_{0}^{t}\cos\omega t\cdot e^{\left(\frac{i_{0}}{BC}t-\frac{U_{0}}{B}\sin\omega t\right)}dt\approx \frac{BC}{{\left(i_{0}\right)}^{2}+{\left(BC\omega \right)}^{2}}e^{\frac{i_{0}}{BC}t}\left[\left(i_{0}\right)\cos\;\omega t+BC\omega sin\;\omega t\right]+D$$
where D is a constant. Substituting Equation (15) into Equation (14), we can have
(16)$$i=\left(A+\frac{Di_{0}U_{0}\omega }{B}\right)e^{-(\frac{i_{0}}{BC}t-\frac{U_{0}}{B}\sin\omega t)}+\frac{i_{0}U_{0}\omega C}{{\left(i_{0}\right)}^{2}+{\left(BC\omega \right)}^{2}}\left[\left(i_{0}\right)\cos\;\omega t+BC\omega sin\omega t\right]e^{\frac{U_{0}}{B}\sin\omega t}$$

Equation (16) can be further simplified as
(17)$$i=\left(A+\frac{Di_{0}U_{0}\omega }{B}\right)e^{-(\frac{i_{0}}{BC}t-\frac{U_{0}}{B}\sin\omega t)}+\frac{i_{0}U_{0}\omega C}{\sqrt{{\left(i_{0}\right)}^{2}+{\left(BC\omega \right)}^{2}}}\left[\sin\left(\omega t+\alpha \right)\right]e^{\frac{U_{0}}{B}\sin\omega t}$$
where
(18)$$\sin\alpha =\frac{i_{0}}{\sqrt{{\left(i_{0}\right)}^{2}+{\left(BC\omega \right)}^{2}}}$$

Substituting Equation (6) into Equation (17), we can have
(19)$$i=\left(A+\frac{Dqi_{0}U_{0}\omega }{kT}\right)e^{-(\frac{qi_{0}}{kTC}t-\frac{qU_{0}}{kT}\sin\omega t)}+\frac{i_{0}U_{0}\omega C}{\sqrt{{\left(i_{0}\right)}^{2}+{\left(\frac{kT}{q}C\omega \right)}^{2}}}\left[\sin\left(\omega t+\alpha \right)\right]e^{\frac{qU_{0}}{kT}\sin\omega t}$$

## 3. Discussion

Equation (19) is the basic equation for NCI–LEDs, where the *i–v* relationship of an LED is approximately simplified to a direct current-driven PN junction. In addition to the above approximations, we make three approximations in solving the above equations. The first approximation is that we ignore the resistance on the circuit. In fact, the resistance on the circuit comes from the wire resistance, which is less than 0.5 Ω. According to the experimental results, the peak current of the circuit is typically less than 10^−5^ A when nanoscale LEDs are used. It can be seen that the applied voltage mainly falls on the NCI–LED. Therefore, we make the first assumption, which is to ignore the resistance of the circuit and the voltage drop across the circuit resistance. If the voltage drop across the circuit resistance is considered in Equation (1), it would be difficult to obtain an analytical solution like Equation (19). Therefore, when analyzing the electrical characteristics of a real NCI–LED device using Equation (19), the current limiting resistor cannot be added to the circuit loop. The second approximation is that we think the current is relatively small. As mentioned above, when using nanoscale LEDs, the peak current of the circuit is less than 10^−5^ A. Therefore, we ignore the *i*^2^ term in Equation (8). This simplifies the differential equation from a nonlinear differential equation to a linear differential equation so that an analytical solution can be obtained. The last approximation is that we only consider the case when the current is stable. This approximation is used to simplify the differential equations to obtain analytical solutions. In the following discussion, it will be found that the calculated results are very close to the experimental results in the low frequency range, which reflects the reasonableness of these approximations.

Equation (19) contains two parts, the first part is an exponential decay term and the second part is a periodic function term with constant amplitude (period: 2π/ω). In other words, the time-varying current of an NCI–LED consists of decay and periodic contributions. The calculated current-time relationship is plotted in Figure 2a, in which the constants *A*, *C*, and *D* are set to −2.5 × 10^−8^, 1 × 10^−13^, and 6 × 10^−3^, respectively. The initial current is relatively large and then decreases exponentially, which corresponds to the exponential decay term in Equation (19). The lifetime of the current (*τ*, defined as when the current decreases to 1/e of the initial state) can be calculated as *kTC*/*qi*_0_. Finally, the current reaches a stable state and the current is periodically changed with a constant amplitude (period: *2π/ω*), which corresponds to the stable term in Equation (19). Typically, only the steady state of the current is observed during experimental measurements, while the exponential decay state is ignored because the duration of the exponential decay state is short and widely used oscilloscopes do not have the capability to record data for long periods of time. 

To verify the correctness of the equations, we prepare an NCI-LED with a structure of Al electrode/Al_2_O_3_ film (50 nm)/LED chip/Al_2_O_3_ film (50 nm)/indium tin oxide electrode. The LED chip has an active area of ~10^−9^ m^2^, and has a commercial structure of u-GaN/n-GaN/multi-quantum wells/p-GaN. The equivalent capacitance in the NCI–LED is about 1 × 10^−13^ F. The wavelength of the electroluminescence center is about 435 nm (inset of Figure 2b, and a maximum light power of ~0.3 μW can be obtained. The current-time characteristics of an NCI–LED is experimentally measured by using a current waveform analyzer (Keysight, CX3324A, Santa Rosa, CA, USA). As shown in Figure 2b, the experimentally measured current characteristics are very close to those obtained by Equation (19), reflecting that Equation (19) can be used to describe the device characteristics.

The exponential decay process in the current can be explained as shown in Figure 3. The NCI–LED is equivalent to an LED sandwiched between two insulators, as schematically illustrated in Figure 3a. When the first forward voltage is applied, positive and negative charges, respectively, accumulate at the electrode/insulator interface, creating a potential drop across the LED. Thus, the electrons and holes inherent in the LED are transported to the MQW for electroluminescence, leaving ionized donor/acceptors on both terminals of the LED. This process corresponds to the charging process of the equivalent capacitor. The number of the transferred charges is defined as Δ*Q*_1_^+^ and the relevant forward current is *i*_1_, as shown in Figure 3b. Then, when the first reverse bias voltage is applied, the amount of accumulated charge at the electrode/insulator interface decreases, corresponding to the discharge process of the capacitor. As the reverse current (*i*_0_) of the LED is much smaller than the forward current (*i*_1_), the accumulated charge does not completely disappear during reverse bias. As a result, there is still charge accumulation on both sides of the insulators, as shown in Figure 3c. The number of the reduced charges is defined as Δ*Q*_1_^−^. Obviously, Δ*Q*_1_^−^ is much smaller than Δ*Q*_1_^+^. As shown in Figure 3d, the second forward bias would charge the capacitors again with a charging current of *i*_2_, causing more charges to accumulate at the electrode/insulator interfaces. As is well known, during the charging process of a capacitor, the charging current decreases as the number of stored charges increases. Thus, *i*_2_ is smaller than *i*_1_, and Δ*Q*_2_^+^ is smaller thanΔ*Q*_1_^+^. Similarly, when the second reversed bias is applied, discharging process results in a slight decrease of accumulated charges, and there are still a large number of charges accumulated on both side of the insulators (Figure 3e). Additionally, the third forward bias would charge the capacitors again with a smaller charging current of *i*_3_ (*i*_3_ < *i*_2_), as shown in Figure 3f. The final result is that the forward current decays exponentially, similar to the exponential decay of the charging current in a capacitor.

This charging-discharging process will finally reach a dynamic equilibrium. That is, the increased accumulated charges (Δ*Q*^+^) during the charging process is equal to the decreased accumulated charges during the discharging process (Δ*Q*^−^), as shown in Figure 3g,h. Thus, the peak value of the forward current keeps unchanged. This physical process corresponds to the second part of Equation (15), the periodic function with constant amplitude. Note that the stable state is more important in practical application because the exponential decay state is instantaneous. Thus, if we only consider the working state of an NCI–LED after a sufficiently long time, we can get the time-variant characteristic of the working current as follows:(20)$$i=\frac{i_{0}U_{0}\omega C}{\sqrt{{\left(i_{0}\right)}^{2}+{\left(\frac{\mathrm{kT}}{q}C\omega \right)}^{2}}}\left[\sin\left(\omega t+\alpha \right)\right]e^{\frac{qU_{0}}{\mathrm{kT}}\sin\omega t}$$

According to Equation (20), the forward current is much higher than the reverse current at steady state, and the width of the forward current pulse is much less than half a period of the applied sinusoidal voltage, as shown in Figure 4a. The reason is that the forward current of a PN junction is much higher than the reversed current. As a result, only one light pulse can be observed in one voltage cycle, which is consistent with the experimental results. As shown in Figure 4b, the experimental current characteristics are very close to those obtained by Equation (20), which also reflects the rationality of these equations.

For NCI–LEDs, it is important to increase the electroluminescence intensity. Simply consider that the electroluminescence intensity is positively correlated with the operating current. Therefore, it is instructive to find a method to increase the working current for improving the electroluminescence intensity. The previous report demonstrates that the electroluminescence intensity is limited due to the low reverse current under the negative bias [4]. Thus, a strategy to increase the electroluminescence density is proposed by employing an anti-parallel LED pair, aiming to increase the reverse current when the LED is working under reversed bias. Here, the Equation (20) gives the mathematical explanation, that is, the increase of *i*_0_ leads to the increase operation current.

According to Equation (20), the current can also be increased by increasing the frequency (*f* = *ω*/2*π*) of the applied sinusoidal voltage, which is consistent with experimental results. The experimental and calculated relationships between peak current and *f* are plotted in Figure 5. Both the experimental and calculated results show that the increase of frequency can dramatically increase the peak current. In the low frequency range (<100 kHz), the experimental data are in perfect agreement with the calculated data. However, in the higher frequency range, the experimental data deviate the calculated data. The reason is that Equations (19) and (20) are obtained by simplifying the LED as a direct current-driven PN junction (Equation (3)), and we do not consider the capacitance associated with the PN junction [17]. As a result, when the NCI–LED operates in the low frequency range, the associated capacitance can be ignored. While when operating in the high frequency range, the effect of capacitance on the *i–v* relationship of a PN junction cannot be ignored, and the *i–v* relationship of the PN junction cannot be described by using Equation (3). Thus, the experimental data deviate from the calculated data in the higher frequency range. However, these results reflect the accuracy of the equations (Equations (19) and (20)) in describing the electrical characteristics of NCI–LEDs under low frequency operation.

Similar to the effect of *ω* on the current, the increase of the equivalent capacitance (*C*) in the NCI–LED can also increase the operating current. Thus, the insulating layer in the NCI–LED should employ a dielectric material with high permittivity and high electric breakdown strength. Additionally, the thickness of the insulating layer should be minimized without causing electrical breakdown of the insulating layer.

Obviously, the current can be increased by increasing the amplitude of the applied voltage. Additionally, according to Equation (20), we can get: (21)$$\ln\left(\frac{i}{U_{0}}\right)\propto U_{0}$$

Thus, we can judge whether the device works in this NCI following our proposed mode by quickly determining if the current–voltage relationship conforms to Equation (20). Figure 6a provide a typical *i–v* curve of an NCI-LED operating at a frequency of 100 kHz. It can be found that the ln(*i*/*U*) is linearly related to *U* (inset of Figure 6a). Thus, we can make the judgement that the electrical performance can be described by Equations (19) and (20), and that the LED plays as a PN junction during the working process. Additionally, we can only consider the unidirectional rectifier characteristics of the LED when performing electrical performance analysis. However, when the applied frequency increases to 5 MHz, the ln(*i*/*U*) is not linearly related to *U* (inset of Figure 6b). Thus, we can make the conclusion that the junction capacitance of the LED plays a key role in the electrical performance. Moreover, we must consider the effect of the junction capacitor and the high-frequency response characteristics of the LED when performing electrical performance analysis.

## 4. Conclusions

In summary, theoretical equations of the time-varying relationship between the current and the voltage of NCI–LEDs are constructed. Mathematically, the time-varying current consists of decaying and periodic contributions, which is well consistent with the experimental result. The proposed theoretical equations are suitable for the analysis of NCI–LEDs when the application frequency is lower than 100 kHz. Key parameters that can improve device performance are discussed, including voltage amplitude, frequency, equivalent capacitance, and LED reverse current. Finally, the ln(*i*/*U*)–*U* relationship is proposed to judge the effect of junction capacitors in LEDs on device performance. The theory presented here can serve as important guidance for the rational design of NCI–LEDs.

## Figures and Tables

**Figure 1 nanomaterials-12-02532-f001:**
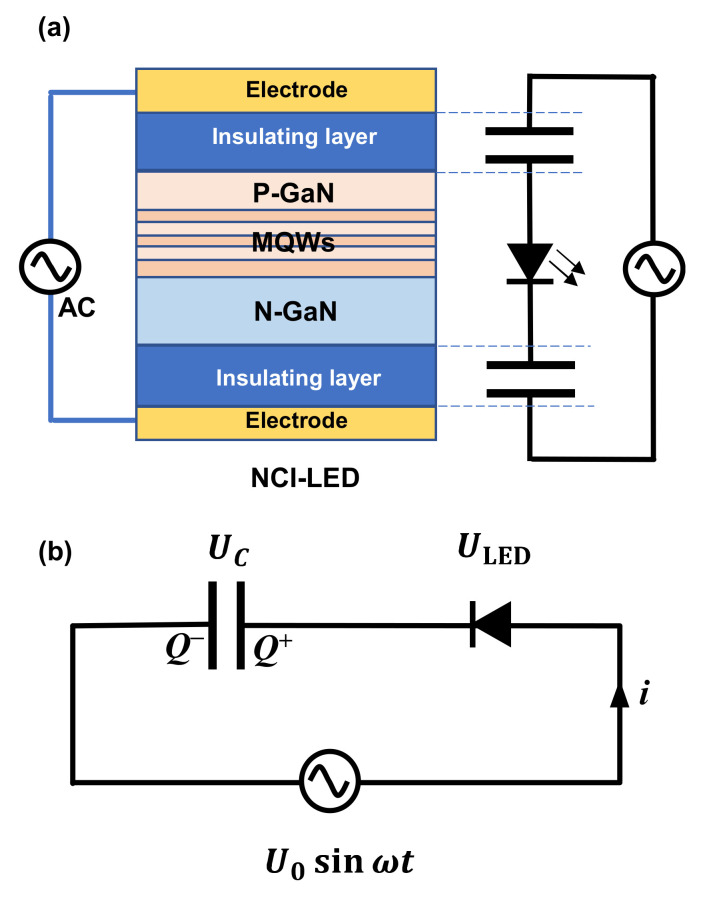
(**a**) Schematic diagram of the NCI-LED and its equivalent circuit diagram. (**b**) The equivalent circuit for derivation of equations.

**Figure 2 nanomaterials-12-02532-f002:**
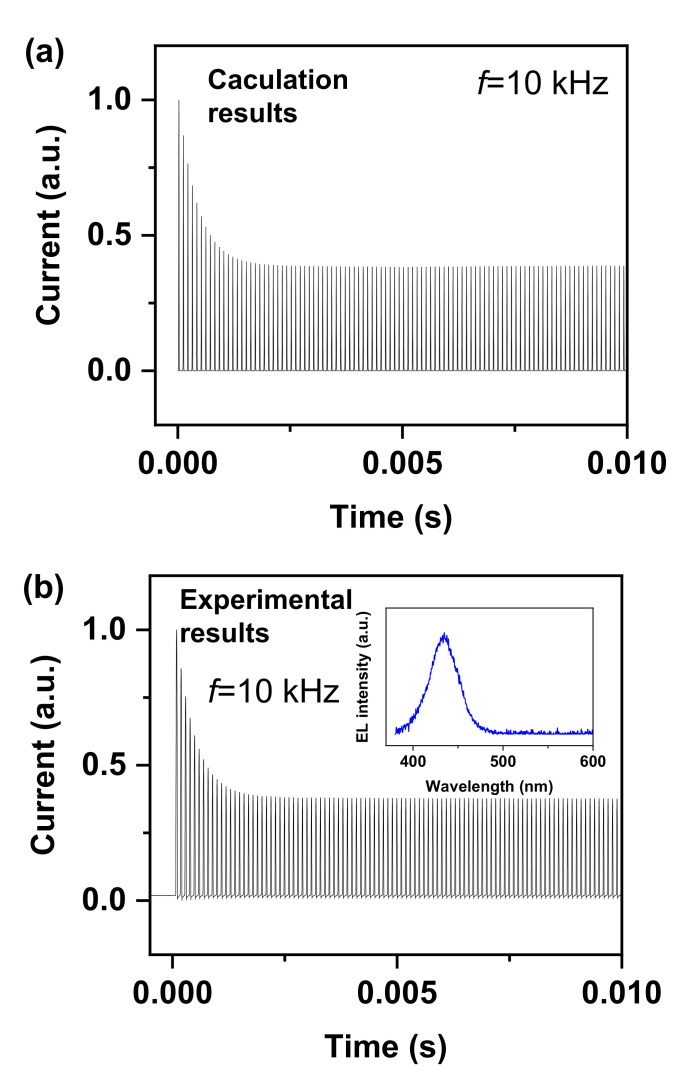
(**a**) Calculation current obtained from Equation (19). (**b**) Experimental time-varying current.

**Figure 3 nanomaterials-12-02532-f003:**
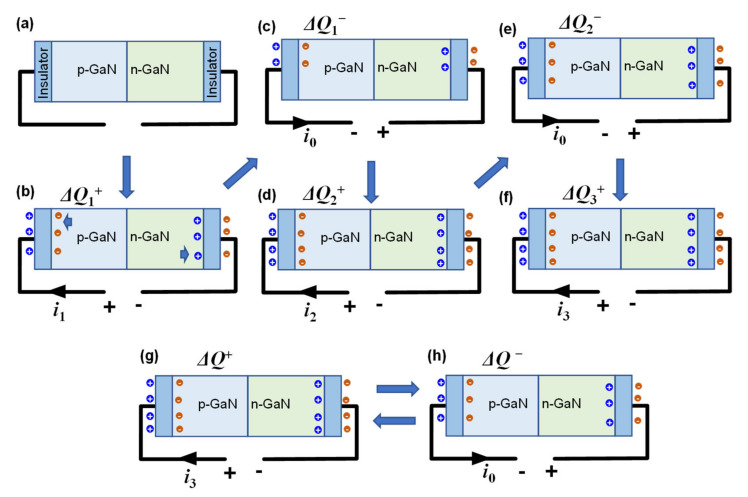
(**a**) Schematic diagram of NCI-LED for working process analysis. (**b**) Device under the first forward bias. (**c**) Device under the first reverse bias. (**d**) Device under the second forward bias. (**e**) Device under the second reverse bias. (**f**) Device under the third forward bias. (**g**) Device under forward bias when the device is operating in the stable state. (**h**) Device under reversed bias when the device is operating in the stable state.

**Figure 4 nanomaterials-12-02532-f004:**
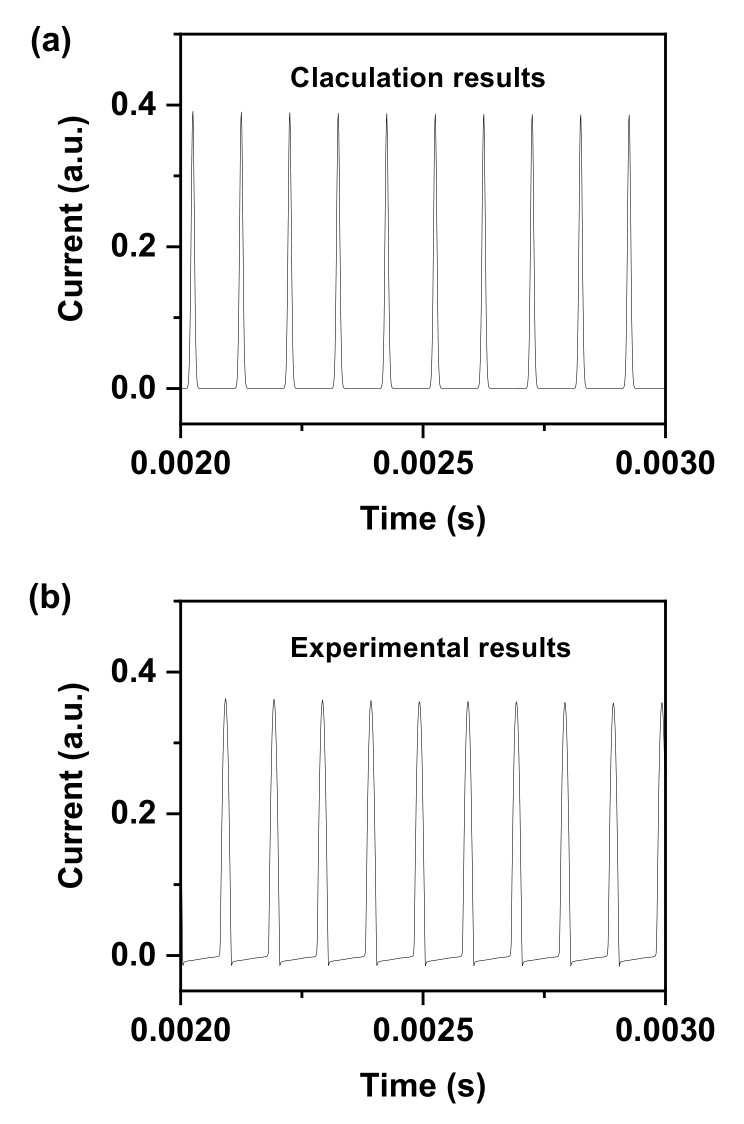
(**a**) Calculation current obtained from Equation (15) when the device in operating in the stable state. (**b**) Experimental time-varying current when the device is operating in the stable state.

**Figure 5 nanomaterials-12-02532-f005:**
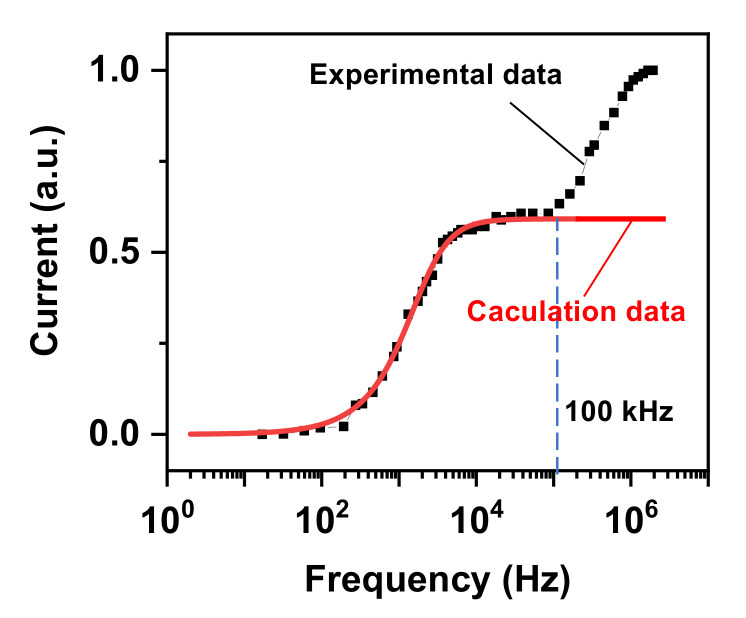
Theoretical and experimental current-frequency relationships.

**Figure 6 nanomaterials-12-02532-f006:**
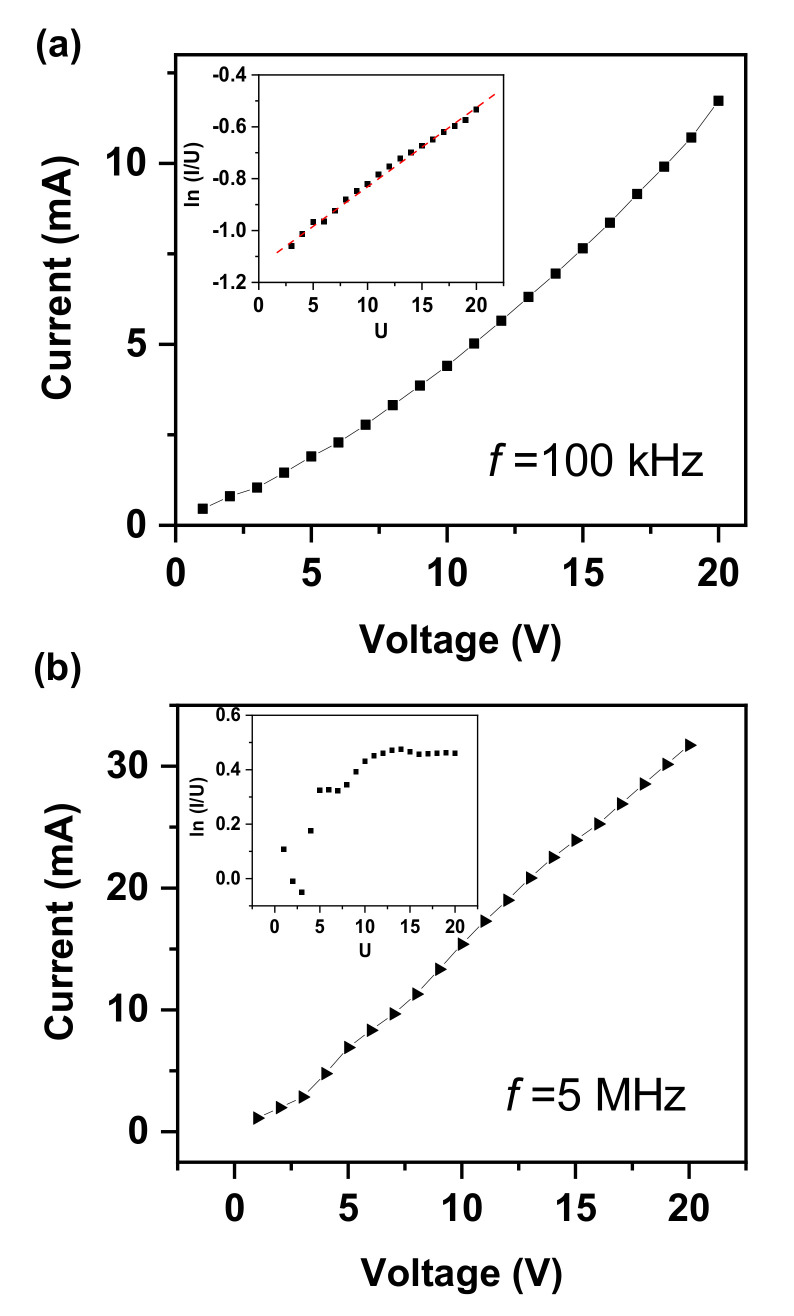
(**a**) *i–v* relationship when the device operating at a frequency of 100 kHz. Inset is the ln(*i*/*U*)-*U* curve. (**b**) *i–v* relationship when the device is operating at a frequency of 5 MHz. Inset is the ln(*i*/*U*)-*U* curve.

## Data Availability

Not applicable.

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
