# Peer review of "Theoretical Study of LED Operating in Noncarrier Injection Mode"

_nanomaterials, 2022, doi:10.3390/nano12152532_

Round 1

Reviewer 1 Report

The authors propose and analyze a model for a nitride LED operating in the non-carrier injection mode. The subject is  interesting and potentially important for applications, it fits the scope of the Nanomaterials journal. However, the presentation of the results needs two substantial improvements:

(1)    The presented model is based on three main approximations (lines 109-110) – ignoring the resistivity of the circuit, assuming that the current is small, considering the stable current situation. The last approximation is analyzed in the further part of the text. I would expect that the first and the second assumption is analyzed in detail. In particular, it is interesting how well the assumed conditions correspond to the real working conditions of the device. How do these assumptions limit the set of devices that can be analyzed with use of the  proposed model? What does it mean “relatively” small current for real devices?

(2)    Fig. 2 shows a comparison of the results of the model calculations with the results of an experiment. However, the description of the experiment and its conditions is extremely limited.  On the other side, there is no information about values for the model parameters (A and D?) used to generate Fig. 2a. Were they fitted to the experimental results?

A few typos should be corrected. In the line 191: “The previous report demonstrates…” a reference to that report should be given.

Reviewer 2 Report

The authors report modeling of the driving  conditions of LED operating in non carrier injection mode. The given electrical equivalent circuit of the LED-NCI is correctly introduced as well as the related formula of the current through the LED-PN junction. Results are of interest to optimize the parameters controling the light emission. I outline to consider the following remarks and questions which are stated below : 

-  Give some details about the the technology of the NCI-LED : materials for electrodes-insulating layers - MQW - typical values of current injection - spectral emission and output light power. 

- There is a correct agreement between experiments and calculation results in Figs 2-4. It is helpful to give the values of the LED parameters which permit to satisfy these conditions in the equivalent circuit. Status on the conditions to optimize LED brightness. 

- In fig 5 experimental datas do not agree with calculation for frequency higher than 100 kHz. Which PN junction capacitance value must be introduced in the model to fit with the experiments and then compare with the experiment. 

To conclude the manuscript outlines results of interest, but as noted above it is required to address and to detail several points more rigorously in a revised form of the paper. 

Reviewer 3 Report

In the article “Theoretical Study of LED Operating in Noncarrier Injection Mode”, the authors investigated the relationship between current and applied voltage for confirming the NCI-LED operating mechanism. The reviewer believes that the author needs to modify the manuscript in minor parts. Therefore, this article can be accepted in Nanomaterials after major revisions.

1.     English writing should be modified. I would suggest the author's review again to correct for awkward phrasing and language. And, please double check the typos.

2.     In Figure 2 and Figure 4, the authors compared the current-time relationship of NCI-LED between experiments and simulations. However, there were little information about experiments. They should describe about experiments more (e.g., measuring condition, fabrication process, etc).

3.     The reviewer believes that the results can be used in practical LED applications. If some high-impact papers are inserted in introduction or conclusion as references, the quality of the paper will be enhanced. The reviewer recommends some state-of-arts and high-impact papers for adding in introduction.

a.     ACS Applied Materials & Interfaces 2022, 14, 24, 28258.

b.     Nature Nanotechnology 16.11 (2021): 1231-1236.

c.      Advanced Optical Materials9(12), 2002211.

Round 2

Reviewer 1 Report

I fully accept all the amendments made by the authors and I find the manuscript suitable for the publication in its present form.

Reviewer 2 Report

The authors in the revised version of the manuscript have correctly considered the questions of the reviewers. Several comments on the characteristics of the LED and their parameter values contribute to improve the interest of the paper 

This version can be accepted for publication. 

Reviewer 3 Report

The authors honestly modified the manuscript, according to the reviewer's comments. The reviewer recommends this manuscript can be published in Nanomaterials in this stage.